# Detection of COVID-19 in smartphone-based breathing recordings: A pre-screening deep learning tool

**Mohanad Alkhodari**📷*, **Ahsan H. Khandoker**

Healthcare Engineering Innovation Center (HEIC), Department of Biomedical Engineering, Khalifa University, Abu Dhabi, UAE

* mohanad.alkhodari@ku.ac.ae

**Data Availability Statement:** Project Coswara COVID-19 dataset is publicly available at https://github.com/iiscleap/Coswara-Data. The shallow and deep breathing datasets included in this study (COVID-19 and healthy subjects) are provided in.

## Abstract

This study was sought to investigate the feasibility of using smartphone-based breathing sounds within a deep learning framework to discriminate between COVID-19, including asymptomatic, and healthy subjects. A total of 480 breathing sounds (240 shallow and 240 deep) were obtained from a publicly available database named Coswara. These sounds were recorded by 120 COVID-19 and 120 healthy subjects via a smartphone microphone through a website application. A deep learning framework was proposed herein that relies on hand-crafted features extracted from the original recordings and from the mel-frequency cepstral coefficients (MFCC) as well as deep-activated features learned by a combination of convolutional neural network and bi-directional long short-term memory units (CNN-BiLSTM). The statistical analysis of patient profiles has shown a significant difference (p-value: 0.041) for ischemic heart disease between COVID-19 and healthy subjects. The Analysis of the normal distribution of the combined MFCC values showed that COVID-19 subjects tended to have a distribution that is skewed more towards the right side of the zero mean (shallow: 0.59±1.74, deep: 0.65±4.35, p-value: <0.001). In addition, the proposed deep learning approach had an overall discrimination accuracy of 94.58% and 92.08% using shallow and deep recordings, respectively. Furthermore, it detected COVID-19 subjects successfully with a maximum sensitivity of 94.21%, specificity of 94.96%, and area under the receiver operating characteristic (AUROC) curves of 0.90. Among the 120 COVID-19 participants, asymptomatic subjects (18 subjects) were successfully detected with 100.00% accuracy using shallow recordings and 88.89% using deep recordings. This study paves the way towards utilizing smartphone-based breathing sounds for the purpose of COVID-19 detection. The observations found in this study were promising to suggest deep learning and smartphone-based breathing sounds as an effective pre-screening tool for COVID-19 alongside the current reverse-transcription polymerase chain reaction (RT-PCR) assay. It can be considered as an early, rapid, easily distributed, time-efficient, and almost no-cost diagnosis technique complying with social distancing restrictions during COVID-19 pandemic.

MAT format at https://github.com/malkhodari/COVID19_breathing_machine_learning. In addition, the development of the trained deep learning model and features extraction codes are all included at the same repository.

**Funding:** This work was supported by a grant (award number: 8474000132) from the Healthcare Engineering Innovation Center (HEIC) at Khalifa University, Abu Dhabi, UAE, and by grant (award number: 29934) from the Department of Education and Knowledge (ADEK), Abu Dhabi, UAE.

**Competing interests:** The authors have declared that no competing interests exist.

## Introduction

Corona virus 2019 (COVID-19), which is a novel pathogen of the severe acute respiratory syndrome coronavirus 2 (SARS-Cov-2), appeared first in late November 2019 and ever since, it has caused a global epidemic problem by spreading all over the world [1]. According to the world heath organization (WHO) April 2021 report [2], there have been nearly 150 million confirmed cases and over 3 million deaths since the pandemic broke out in 2019. Additionally, the United States (US) have reported the highest number of cumulative cases and deaths with over 32.5 million and 500,000, respectively. These huge numbers have caused many healthcare services to be severely burdened especially with the ability of the virus to develop more genomic variants and spread more readily among people. India, which is one of the world's biggest suppliers of vaccines, is now severely suffering from the pandemic after the explosion of cases due to a new variant of COVID-19. It has reached more than 17.5 million confirmed cases, setting it behind the US as the second worst hit country [2, 3].

COVID-19 patients usually range from being asymptomatic to developing pneumonia and in severe cases, death. In most reported cases, the virus remains incubation for a period of 1 to 14 days before the symptoms of an infection start arising [4]. Patients carrying COVID-19 have exhibited common signs and symptoms including cough, shortness of breath, fever, fatigue, and other acute respiratory distress syndromes (ARDS) [5, 6]. Most infected people suffer from mild to moderate viral symptoms, however, they end up by being recovered. On the other hand, patients who develop severe symptoms such as severe pneumonia are mostly people over 60 years of age with conditions such as diabetes, cardiovascular diseases (CVD), hypertension, and cancer [4, 5]. On most cases, the early diagnosis of COVID-19 helps in preventing its spreading and development to severe infection stages. This is usually done by following steps of early patient isolation and contact tracing. Furthermore, timely medication and efficient treatment reduces symptoms and results in lowering the mortality rate of this pandemic [7].

The current gold standard in diagnosing COVID-19 is the reverse-transcription polymerase chain reaction (RT-PCR) assay [8, 9]. It is the most commonly used technique worldwide to successfully confirm the existence of this viral infection. Additionally, examinations of the ribonucleic acid (RNA) in patients carrying the virus provide further information about the infection, however, it requires longer time for diagnosis and is not considered as accurate as other diagnostic techniques [10]. The integration of computed tomography (CT) screening (X-ray radiations) is another effective diagnostic tool (sensitivity $\geq$90%) that often provides supplemental information about the severity and progression of COVID-19 in lungs [11, 12]. CT imaging is not recommended for patients at the early stages of the infection, i.e., showing asymptomatic to mild symptoms. It provides useful details about the lungs in patients with moderate to severe stages due to the disturbances in pulmonary tissues and its corresponding functions [13].

Most recently, several studies have utilized the new emerging algorithms in artificial intelligence (AI) to detect and classify COVID-19 in CT and X-ray images [14]. Machine and deep learning algorithms were implemented in several studies (taking CT images as inputs) with a discrimination accuracy reaching over 95% between healthy and infected subjects [15–20]. The major contribution of these studies is the ability of trained models including support vector machine (SVM) and convolutional neural networks (CNN) in detecting COVID-19 in CT images with minimal pre-processing steps. Moreover, several studies have utilized deep learning with extra feature fusion techniques and entropy-controlled optimization [21], rank-based average pooling [22], pseudo-Zernike moment (PZM) [23], and internet-of-things [24] to detect COVID-19 in CT images. In addition, there has been extensive research carried out for COVID-19 assessment using X-ray images and machine learning [25–27]. Majority of the current state-of-art approaches rely on two-dimensional (2D) X-ray images of the lungs to train

neural networks on extracting features and thus identifying subjects carrying the viral infection. Despite of the high levels of accuracy achieved in most of the studies, CT and X-ray imaging use ionizing radiations that make them not feasible for frequent testing. In addition, these imaging modalities may not be available in all public healthcare services, especially for countries who are swamped with the pandemic, due to their costs and additional maintenance requirements. Most recently, researchers have utilized a safer and simpler imaging approach based on ultrasound to screen lungs for COVID-19 [28] and achieved high levels of performance (accuracy > 89%). Therefore, finding promising alternatives that are simple, fast, and cost-effective is an ultimate goal to researchers when it comes to integrating these techniques with machine learning.

Biological respiratory signals, such as coughing and breathing sounds, could be another promising tool to indicate the existence of the viral infection [29], as these signals have a direct connection with lungs. Respiratory auscultation is considered as a safe and non-invasive technique to diagnose the respiratory system and its associated organs. This technique is usually done by clinicians using an electronic stethoscope to hear and record the air sound moving inside and outside lungs while breathing or coughing. Thus, an indication of any pulmonary anomalies could be detected and identified [30–32]. Due to the simplicity in recording respiratory signals, lung sounds could carry useful information about the viral infection, and thus, could set an early alert to the patient before moving on with further and extensive medication procedures. In addition, combining the simple respiratory signals with AI algorithms could be a key to enhance the sensitivity of detection for positive cases due to its ability to generalize over a wide set of data with less computational complexity [33].

Many studies have investigated the information carried by respiratory sounds in patients tested positive for COVID-19 [34–36]. Furthermore, it has been found that vocal patterns extracted from COVID-19 patients' speech recordings carry indicative biomarkers for the existence of the viral infection [37]. In addition, a telemedicine approach was also explored to observe evidences on the sequential changes in respiratory sounds as a result of COVID-19 infection [38]. Most recently, AI was utilized in one study to recognize COVID-19 in cough signals [39] and in another to evaluate the severity of patients' illness, sleep quality, fatigue, and anxiety through speech recordings [40]. Despite of the high levels of performance achieved in the aforementioned AI-based studies, further investigations on the capability of respiratory sounds in carrying useful information about COVID-19 are still required, especially when embedded within the framework of sophisticated AI-based algorithms. Furthermore, due to the explosion in the number of confirmed positive COVID-19 cases all over the world, it is essential to ensure providing a system capable of recognizing the disease in signals recording through portable devices, such as computers or smartphones, instead of regular clinic-based electronic stethoscopes.

Motivated by the aforementioned, a complete deep learning approach is proposed in this paper for a successful detection of COVID-19 using only breathing sounds recorded through a microphone of a smartphone device (Fig 1). The proposed approach serves as a rapid, no-cost, and easily distributed pre-screening tool for COVID-19, especially for countries who are in a complete lockdown due to the wide spread of the pandemic. Although the current gold standard, RT-PCR, provides high success rates in detecting the viral infection, it has various limitations including the high expenses involved with equipment and chemical agents, requirement of expert nurses and doctors for diagnosis, violation of social distancing, and the long testing time required to obtain results (2-3 days). Thus, the development of a deep learning model overcomes most of these limitations and allows for a better revival in the healthcare and economic sectors in several countries.

Furthermore, the novelty of this work lies in utilizing smartphone-based breathing recordings within this deep learning model, which, when compared to conventional respiratory

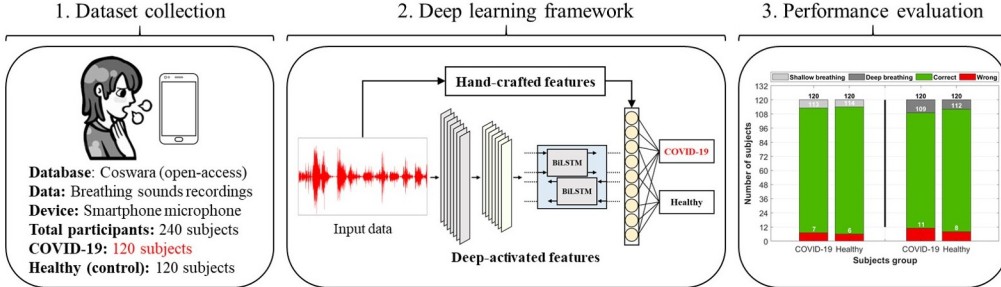

**Fig 1. A graphical abstract of the complete procedure followed in this study.** The input data includes breathing sounds collected from an open-access database for respiratory sounds (Coswara [41]) recorded via smartphone microphone. The data includes a total of 240 participants, out of which 120 subjects were suffering from COVID-19, while the remaining 120 were healthy (control group). A deep learning framework was then utilized based on hand-crafted features extracted by feature engineering techniques, as well as deep-activated features extracted by a combination of convolutional and recurrent neural network. The performance was then evaluated and further discussed on the use of artificial intelligence (AI) as a successful pre-screening tool for COVID-19.

auscultation devices, i.e., electronic stethoscopes, are more preferable due to their higher accessibility by wider population. This plays an important factor in obtaining medical information about COVID-19 patients in a timely manner while at the same time maintaining an isolated behaviour between people. Additionally, this study covers patients who are mostly from India, which is severely suffering from a new genomic variant (first reported in December 2020) of COVID-19 capable of escaping the immune system and most of the available vaccines [2, 42]. Thus, it gives an insight on the ability of AI algorithms in detecting this viral infection in patients carrying this new variant, including asymptomatic.

Lastly, the study presented herein investigates signal characteristics contaminated within shallow and deep breathing sounds of COVID-19 and healthy subjects through deep-activated attributes (neural network activations) of the original signals as well as wide attributes (hand-crafted features) of the signals and their corresponding mel-frequency cepstrum (MFC). The utilization of one-dimensional (1D) signals within a successful deep learning framework allows for a simple, yet effective, AI design that does not require heavy memory requirements. This serves as a suitable solution for further development of telemedicine and smartphone applications for COVID-19 (or other pandemics) that can provide real-time results and communications between patients and clinicians in an efficient and timely manner. Therefore, as a pre-screening tool for COVID-19, this allows for a better and faster isolation and contact tracing than currently available techniques.

## Materials and methods

### Dataset collection and subjects information

The dataset used in this study was obtained from Coswara [41], which is a project aiming towards providing an open-access database for respiratory sounds of healthy and unhealthy individuals, including those suffering from COVID-19. The project is a worldwide respiratory data collection effort that was first initiated in August, 7th 2020. Ever since, it has collected data from more than 1,600 participants (Male: 1185, Female: 415) from allover the world (mostly Indian population). The database was approved by the Indian institute of science (IISc), human ethics committee, Bangalore, India, and conforms to the ethical principles outlined in the declaration of Helsinki. No personally identifiable information about participants was collected and the participants' data was fully anonymized during storage in the database.

The database includes breath, cough, and voice sounds acquired via crowdsourcing using an interactive website application that was built for smartphone devices [43]. The average interaction time with the application was 5-7 minutes. All sounds were recorded using the microphone of a smartphone and sampled with a sampling frequency of 48 kHz. The participants had the freedom to select any device for recording their respiratory sounds, which reduces device-specific bias in the data. The audio samples (stored in. WAV format) for all participants were manually curated through a web interface that allows multiple annotators to go through each audio file and verify the quality as well as the correctness of labeling. All participants were requested to keep a 10 cm distance between the face and the device before starting the recording.

So far, the database had a COVID-19 participants' count of 120, which is almost 1-10 ratio to healthy (control) participants. In this study, all COVID-19 participants' data was used, and the same number of samples from the control participants' data was randomly selected to ensure a balanced dataset. Therefore, the dataset used in this study had a total of 240 subjects (COVID-19: 120, Control: 120). Furthermore, only breathing sounds of two types, namely shallow and deep, were obtained from every subject and used for further analysis. Figs 2 and 3 show examples from the shallow and deep breathing datasets, respectively, with their corresponding spectrogram representation. To ensure the inclusion of maximum information from each breathing recording as well as to cover at least 2-4 breathing cycles (inhale and exhale), a total of 16 seconds were considered, as the normal breathing pattern in adults ranges between 12 to 18 breaths per minute [44]. All recordings less than 16 seconds were padded with zeros. Furthermore, the final signals were resampled with a sampling frequency of 4 kHz.

The demographic and clinical information of the selected subjects is provided in Table 1. All values are provided as range and mean±std (age), numbers (sex), and yes/no (1/0). To check for most significant variables, a linear regression fitting algorithm [45] was applied. An

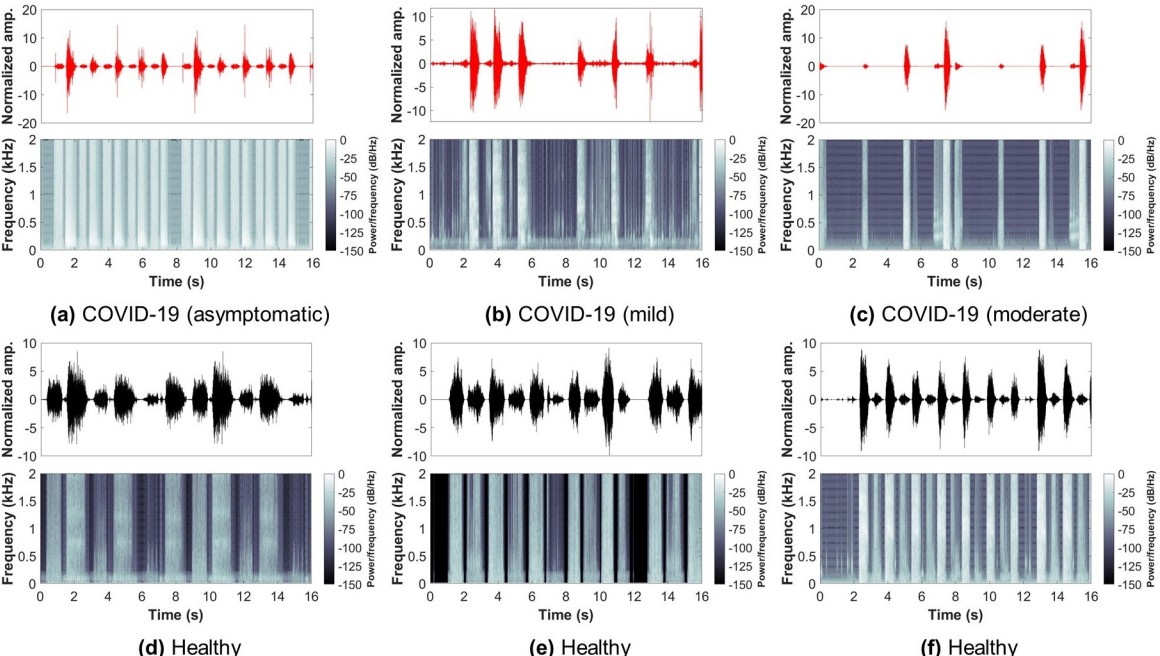

**Fig 2. Examples from the shallow breathing sounds recorded via smartphone microphone along with their corresponding spectrograms.** Showing: (a-c) COVID-19 subjects (asymptomatic, mild, moderate), (d-f) healthy subjects.

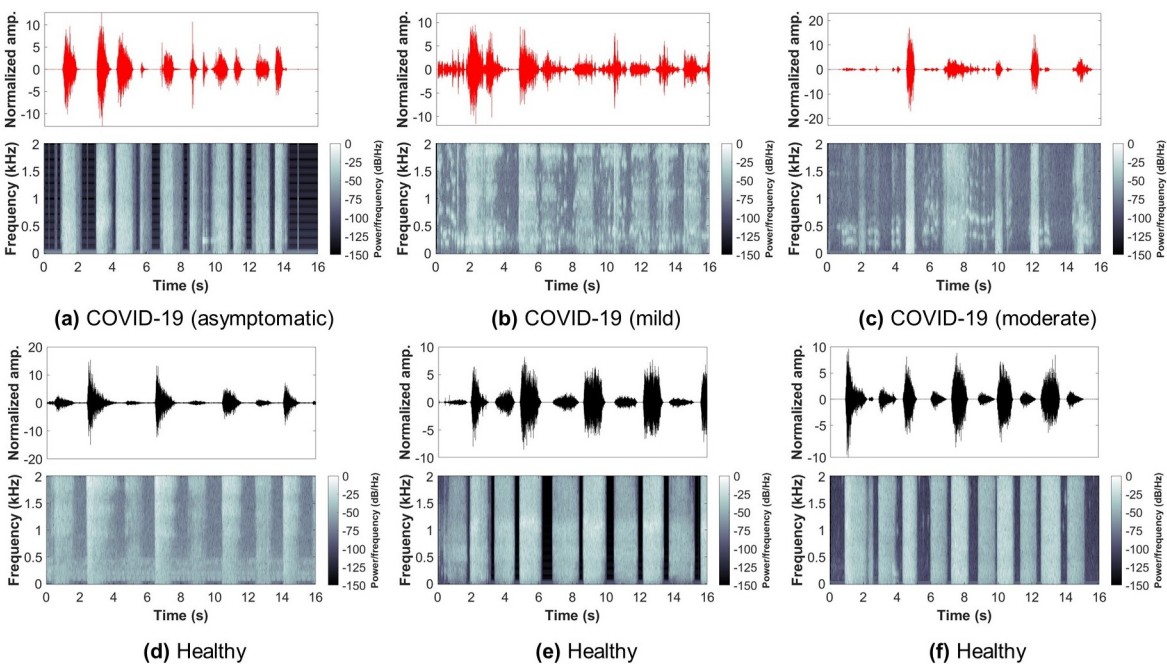

**Fig 3. Examples from the deep breathing sounds recorded via smartphone microphone along with their corresponding spectrograms.** Showing: (a-c) COVID-19 subjects (asymptomatic, mild, moderate), (d-f) healthy subjects.

**Table 1. The demographic and clinical information of COVID-19 and healthy (control) subjects included in the study.**

| Category | COVID-19 | | | | Healthy (Control) | p-value |
|---|---|---|---|---|---|---|
| | Asymptomatic | Mild | Moderate | Overall | | |
| **Demographic information** | | | | | | |
| Number of subjects | 20 | 90 | 10 | 120 | 120 | - |
| Age (Mean±Std) | 20-77 (32.65±13.69) | 15-70 (33.43±12.99) | 23-65 (43.33±15.46) | 15-77 (34.04±13.45) | 15-70 (36.02±13.06) | 0.149 |
| Sex (Male / Female) | 10 / 10 | 65 / 25 | 7 / 3 | 82 / 38 | 85/35 | 0.612 |
| **Comorbidities** | | | | | | |
| Diabetes | 1 | 7 | 1 | 9 | 11 | 0.629 |
| Hypertension | 0 | 6 | 1 | 7 | 6 | 0.788 |
| Chronic lung disease | 1 | 1 | 0 | 2 | 0 | 0.159 |
| Ischemic heart disease | 1 | 3 | 0 | 4 | 0 | **0.041** |
| Pneumonia | 0 | 3 | 0 | 3 | 0 | 0.083 |
| **Health conditions** | | | | | | |
| Fever | 0 | 39 | 6 | 45 | 1 | **<0.001** |
| Cold | 0 | 37 | 4 | 41 | 6 | **<0.001** |
| Cough | 0 | 44 | 4 | 48 | 13 | **<0.001** |
| Muscle pain | 2 | 15 | 5 | 22 | 1 | **<0.001** |
| Loss of smell | 0 | 15 | 3 | 18 | 0 | **<0.001** |
| Sore throat | 0 | 26 | 3 | 29 | 2 | **<0.001** |
| Fatigue | 1 | 18 | 3 | 22 | 1 | **<0.001** |
| Breathing Difficulties | 1 | 7 | 6 | 14 | 0 | **<0.001** |
| Diarrhoea | 0 | 1 | 0 | 1 | 0 | 0.159 |

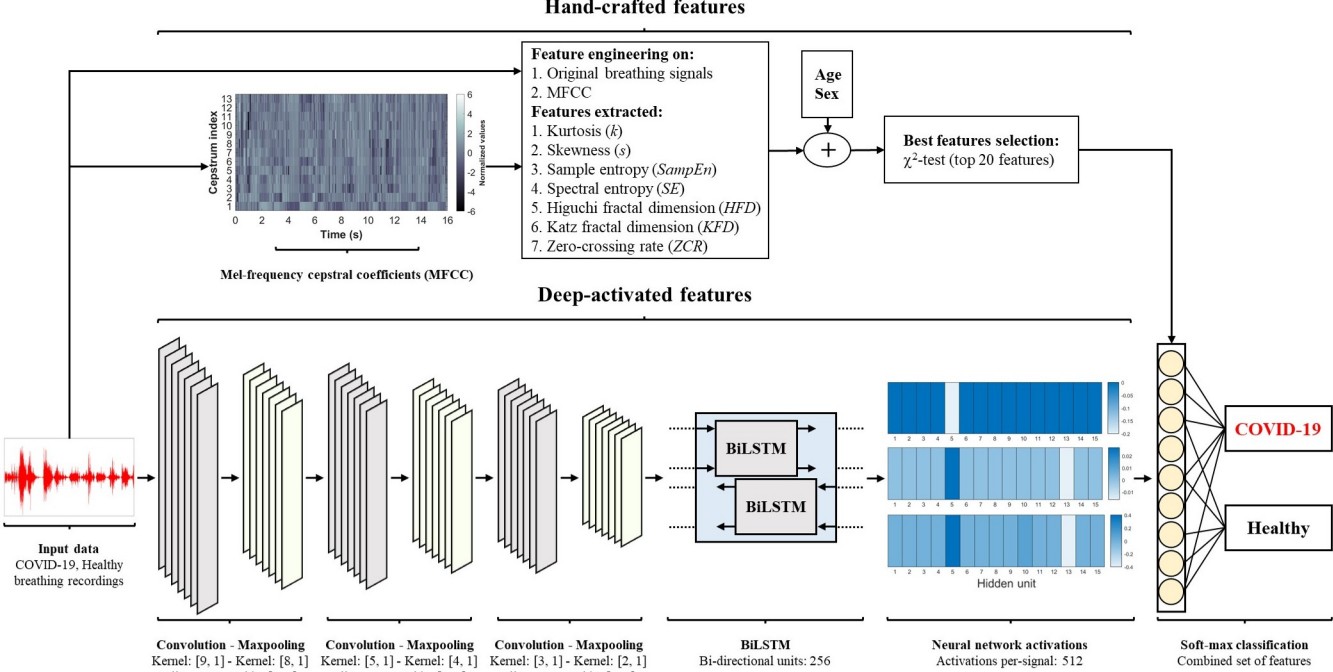

**Fig 4. The framework of deep learning followed in this study.** The framework includes a combination of hand-crafted features and deep-activated features. Deep features were obtained through a combined convolutional and recurrent neural network (CNN-BiLSTM), and the final classification layer uses both features sets to discriminate between COVID-19 and healthy subjects.

indication of a significant difference between COVID-19 and healthy subjects was obtained whenever the p-value was less than 0.05 (bold).

## Deep learning framework

The deep learning framework proposed in this study (Fig 4) includes a combination of hand-crafted features as well as deep-activated features learned through model's training and reflected as time-activations of the input. To extract hand-crafted features, various algorithm and functions were used to obtain signal attributes from the original breathing recording and from its corresponding mel-frequency cepstral coefficients (MFCC). In addition, deep-activated learned features were obtained from the original breathing recording through a combined neural network that consists of convolutional and recurrent neural networks. Each part of this framework is briefly described in the following subsections and illustrated as a pseudo-code in 1.

**Algorithm 1** Training Deep Learning Model for COVID-19 Prediction

```
Input: 120 COVID-19 / 120 healthy breathing recordings (shallow or
       deep)
Output: Trained model to predict COVID-19 or healthy
1: for Every breathing recording do
2:    Calculate kurtosis k (Eq 1) and skewness s (Eq 2)
3:    Calculate eample entropy SampEn (Eq 3) and spectral entropy SE
      (Eq 4)
4:    Calculate fractal dimensions—Higuchi HFD (Eq 5) and Katz KFD (Eq
      6)
5:    Calculate zero-crossing rate ZCR (Eq 7)
6:    Extract 13 Mel-frequency cepstral coefficients (MFCC)
```

```
7:   for Every MFCC signal do
8: Extract all aforementioned hand-crafted features
9:   end for
10: end for
11: χ²-test: All features plus age and sex, select best 20 features
12: for Every breathing recording do
13:   Apply volume control and time shift augmentation (24 augmented
signals)
14: end for
15: Train CNN-BiLSTM network to extract deep-activated features from
breathing recordings
16: Update the trained model using the 20 best hand-crafted features
```

**Hand-crafted features.** These features refer to signal attributes that are extracted manually through various algorithms and functions in a process called feature engineering [46]. The advantage of following such process is that it can extract internal and hidden information within input data, i.e., sounds, and represent it as single or multiple values [47]. Thus, additional knowledge about the input data can be obtained and used for further analysis and evaluation. Hand-crafted features were initially extracted from the original breathing recordings, then, they were also extracted from the MFCC transformation of the signals. The features included in this study are,

*Kurtosis and skewness*: In statistics, kurtosis is a quantification measure for the degree of extremity included within the tails of a distribution relative to the tails of a normal distribution. The more the distribution is outlier-prone, the higher the kurtosis values, and vice-versa. A kurtosis of 3 indicates that the values follow a normal distribution. On the other hand, skewness is a measure for the asymmetry of the data that deviates it from the mean of the normal distribution. If the skewness is negative, then the data are more spread towards the left side of the mean, while a positive skewness indicates data spreading towards the right side of the mean [48]. A skewness of zero indicates that the values follow a normal distribution. Kurtosis ($k$) and skewness ($s$) can be calculated as,

$$k = E\left[\frac{(X - \mu)^4}{\sigma^4}\right] \tag{1}$$

$$s = E\left[\frac{(X - \mu)^3}{\sigma^3}\right] \tag{2}$$

where $X$ included input values, $\mu$ and $\sigma$ are the mean and standard deviation values of the input, respectively, and $E$ is an expectation operator.

*Sample entropy*: In physiological signals, the sample entropy (SampEn) provides a measure for complexity contaminated within time sequences. This feature represent the randomness contaminated within a signal by embedding it into a phase space to estimate the increment rate in the number of phase space patterns. It can be calculated though the negative natural logarithm of a probability that segments of length $m$ match their consecutive segments under a value of tolerance ($r$) [49] as follows,

$$SampEn = -log\left(\frac{segment_A}{segment_{A+1}}\right) \tag{3}$$

where $segment_A$ is the first segment in the time sequence and $segment_{A+1}$ is the consecutive segment.

*Spectral entropy*: To measure time series irregularity, spectral entropy (SE) provides a frequency domain entropy measure as a sum of the normalize signal spectral power [50]. It differs from the aforementioned SampEN in analyzing the frequency spectrum of signals rather than time sequences and phase. Based on Shannon's entropy, the SE can be calculated as,

$$SE = -\sum_{n=1}^{N} P(n) \times log(P(n)) \tag{4}$$

where $N$ is the total number of frequency points and $P(n)$ is the probability distribution of the power spectrum.

*Fractal dimension*. Higuchi and Katz [51, 52] provided two methods to measure statistically the complexity in a time series. More specifically, fractal dimension measures provide an index for characterizing how much a time series is self-similar over some region of space. Higuchi (*HFD*) and Katz (*KFD*) fractal dimensions can be calculated as,

$$HFD = \frac{log(L(r))}{log(1/r)} \tag{5}$$

$$KFD = \frac{log(N)}{log(N) + log(d/L(r))} \tag{6}$$

where $L(k)$ is the length of the fractal curve, $r$ is the selected time interval, $N$ is the length of the signal, and $d$ is the maximum distance between an initial point to other points.

*Zero-crossing rate*. To measure the number of times a signal has passed through the zero point, a zero-crossing rate (ZCR) measure is provided. In other words, ZCR refers to the rate of sign-changes in the signals' data points. It can be calculated as follows,

$$ZCR = \frac{1}{T}\sum_{t=1}^{T}(|x_t - x_{t+1}|) \tag{7}$$

where $x_t = 1$ if the signal has a positive value at time step $t$ and a value of 0 otherwise.

*Mel-frequency cepstral coefficients (MFCC)*. To better represent speech and voice signals, MFCC provides a set of coefficients of the discrete cosine transformed (DCT) logarithm of a signal's spectrum (mel-frequency cepstrum (MFC)). It is considered as an overall representation of the information contaminated within signals regarding the changes in its different spectrum bands [53, 54]. Briefly, to obtain the coefficients, the signals goes through several steps, namely windowing the signal, applying discrete Fourier transform (DFT), calculating the log energy of the magnitude, transforming the frequencies to the Mel-scale, and applying inverse DCT.

In this work, 13 coefficients (MFCC-1 to MFCC-13) were obtained from each breathing sound signal. For every coefficient, the aforementioned features were extracted and stored as an additional MFCC hand-crafted features alongside the original breathing signals features.

**Deep-activated features.** These features refer to attributes extracted from signals through a deep learning process and not by manual feature engineering techniques. The utilization of deep learning allows for the acquisition of optimized features extracted through deep convolutional layers about the structural information contaminated within signals. Furthermore, it has the ability to acquire the temporal (time changes) information carried through time sequences [55–57]. Such optimized features can be considered as a complete representation of the input data generated iteratively through an automated learning process. To achieve this, we used an

advanced neural network based on a combination of convolutional neural network and bi-directional long short-term memory (CNN-BiLSTM).

*Neural network architecture.* The structure of the network starts by 1D convolutional layers. In deep learning, convolutions refer to a multiple number of dot products applied to 1D signals on pre-defined segments. By applying consecutive convolutions, the network extracts deep attributes (activations) to form an overall feature map for the input data [56]. A single convolution on an input $x_i^0 = [x_1, x_2, ..., x_n]$, where $n$ is the total number of points, is usually calculated as,

$$c_i^{lj} = h(b_j + \sum_{m=1}^{M} w_m^j x_{i+m-1}^j) \tag{8}$$

where $l$ is the layer index, $h$ is the activation function, $b$ is the bias of the $j^{th}$ feature map, $M$ is the kernel size, $w_m^j$ is the weight of the $j^{th}$ feature map and $m^{th}$ filter index.

In this work, three convolutional layers were used to form the first stage of the deep neural network. The kernel sizes of each layer are [9, 1], [5, 1], and [3, 1], respectively. Furthermore, the number of filters increases as the network becomes deeper, that is 16, 32, and 64, respectively. Each convolutional layer was followed by a max-pooling layer to reduce the dimensionality as well as the complexity in the model. The max-pooling kernel size decreases as the network gets deeper with a [8, 1], [4, 1], and [2, 1] kernels for the three max-pooling layers, respectively. It is worth noting that each max-pooling layer was followed by a batch normalization (BN) layer to normalize all filters as well as by a rectified linear unit (ReLU) layer to set all values less than zero in the feature map to zero. The complete structure is illustrated in Fig 4.

The network continues with additional extraction of temporal features through bi-directional LSTM units. In recurrent neural networks, LSTM units allows for the detection of long short-term dependencies between time sequence data points. Thus, it overcomes the issues of exploding and vanishing gradients in chain-like structures during training [55, 58]. An LSTM block includes a collection of gates, namely input ($i$), output ($o$), and forget ($f$) gates. These gates handle the flow of data as well as the processing of the input and output activations within the network's memory. The information of the main cell ($C_t$) at any instance ($t$) within the block can be calculated as,

$$C_t = f_t C_{t-1} + i_t c_t \tag{9}$$

where $c_t$ is the input to the main cell and $C_{t-1}$ includes the information at the previous time instance.

In addition, the network performs hidden units ($h_t$) activations on the output and main cell input using a sigmoid function as follows,

$$h_t = o_t \sigma(c_t) \tag{10}$$

Furthermore, a bi-drectional functionality (BiLSTM) allows the network to process data in both the forward and backward direction as follows,

$$y_t = W_{\overrightarrow{h}y} \overrightarrow{h^N} + W_{\overleftarrow{h}y} \overleftarrow{h^N} + b_y \tag{11}$$

where $\overrightarrow{h^N}$ and $\overleftarrow{h^N}$ are the outputs of the hidden layers in the forward and backward directions, respectively, for all $N$ levels of stack and $b_y$ is a bias vector.

In this work, a BiLSTM hidden units functionality was selected with a total number of hidden units of 256. Thus, the resulting output is a 512 vector (both directions) of the extracted hidden units of every input.

*BiLSTM activations*. To be able to utilize the parameters that the BiLSTM units have learned, the activations that correspond to each hidden unit were extracted from the network for each input signal. Recurrent neural network activations of a pre-trained network are vectors that carry the final learned attributes about different time steps within the input [59, 60]. In this work, these activations were the final signal attributes extracted from each input signal. Such attributes are referred to as deep-activated features in this work (Fig 4). To compute these activations, we use function *activations() in MATLAB inputting the trained network, the selected data (breathing recording), and the chosen features layer (BiLSTM)*. Furthermore, they were concatenated with the hand-crafted features alongside age and sex information and used for the final predictions by the network.

**Network configuration and training scheme.** Prior to deep learning model training, several data preparation and network fine-tuning steps were followed including data augmentation, best features selection, deciding the training and testing scheme, and network parameters configuration.

*Data augmentation*. Due to the small sample size available, it is critical for deep learning applications to include augmented data. Instead of training the model on the existing dataset only, data augmentation allows for the generation of new modified copies of the original samples. These new copies have similar characteristics of the original data, however, they are slightly adjusted as if they are coming from a new source (subject). Such procedure is essential to expose the deep learning model to more variations in the training data. Thus, making it robust and less biased when attempting to generalize the parameters on new data [61]. Furthermore, it was essential to prevent the model from over-fitting, where the model learns exactly the input data only with a very minimal generalization capabilities for unseen data [62].

In this study, 3,000 samples per class were generated using two 1D data augmentation techniques as follows,

- *Volume control*: Adjusts the strength of signals in decibels (dB) for the generated data [63] with a probability of 0.8 and gain ranging between -5 and 5 dB.

- *Time shift*: Modifies time steps of the signals to illustrate shifting in time for the generated data [64] with a shifting range of [-0.005 to 0.005] seconds.

*Best features selection*. To ensure the inclusion of the most important hand-crafted features within the trained model, a statistical univariate chi-square test ($\chi^2$-test) was applied. In this test, a feature is decided to be important if the observed statistical analysis using this feature matches with the expected one, i.e., label [65]. Furthermore, an important feature indicates that it is considered significant in discriminating between two categories with a $p$-value $< 0.05$. The lower the $p$-value, the more the feature is dependent on the category label. The importance score can then be calculated as,

$$score = -log(p) \qquad (12)$$

In this work, hand-crafted features extracted from the original breathing signals and from the MFCC alongside the age and sex information were selected for this test. The best 20 features were included in the final best features vector within the final fully-connected layer (along with the deep-activated features) for predictions.

*Training configuration*. To ensure the inclusion of the whole available data, a leave-one-out training and testing scheme was followed. In this scheme, a total of 240 iterations (number of input samples) were applied, where in each iteration, an $i^{th}$ subject was used as the testing subject, and the remaining subjects were used for model's training. This scheme was essential to be followed to provide a prediction for each subject in the dataset.

Furthermore, the network was optimized using adaptive moment estimation (ADAM) solver [66] and with a learning rate of 0.001. The L2-regularization was set to $10^6$ and the mini-batch size to 32.

## Performance evaluation

The performance of the proposed deep learning model in discriminating COVID-19 from healthy subjects was evaluated using traditional evaluation metrics including accuracy, sensitivity, specificity, precision, and F1-score. Additionally, the area under the receiver operating characteristic (AUROC) curves was analysed for each category to show the true positive rate (TPR) versus the false positive rate (FPR) [67].

## Results

### Patient clinical information

COVID-19 subjects included in this study had an average age of 34.04 years (± 13.45), while healthy subjects were slightly higher with an average of 36.02 years(±13.06). However, no significant difference was obtained for this variable across subjects (p = 0.149). It is worth noting that COVID-19 subjects with moderate conditions had a higher average age of 43.33 years. The distribution of male/female subjects across the two categories was close to 2:1 ratio with a majority of male subjects. Sex was not significantly different between COVID-19 and healthy subjects (p = 0.612).

Comorbidities including diabetes, hypertension, chronic lung disease, and pneumonia were not found significantly different between COVID-19 and healthy subjects. However, the majority of COVID-19 subjects suffering from these diseases were in the mild group. The only important variable was the ischemic heart disease with a p-value of 0.041. Only 4 subjects were suffering from disease while having COVID-19, while no healthy subjects were recorded with this disease. All health conditions were found significantly different (p < 0.001) between COVID-19 and healthy subjects except for diarrhoea (p = 0.159). Significant health conditions included in this dataset were fever, cold, cough, muscle pain, loss of smell, sore throat, fatigue, and breathing difficulties. It is worth noting that only 4 subjects in the asymptomatic COVID-19 group were suffering from muscle pain, fatigue, and breathing difficulties.

### Analysis of MFCC

Examples of the 13 MFCC extracted from the original shallow and deep breathing signals are illustrated in Figs 5 and 6, respectively, for COVID-19 and healthy subjects. Furthermore, the figures show MFCC values (after summing all coefficients) distributed as a normal distribution. From the figure, the normal distribution of COVID-19 subjects was slightly skewed to the right side of the mean, while the normal distribution of the healthy subjects was more towards the zero mean, indicating that it is better in representing a normal distribution. It is worth noting that the MFCC values of shallow breathing were lower than deep breathing in both COVID-19 and healthy subjects.

Tables 2 and 3 show the values of the combined MFCC values, kurtosis, and skewness among all COVID-19 and healthy subjects (mean±std) for the shallow and deep breathing

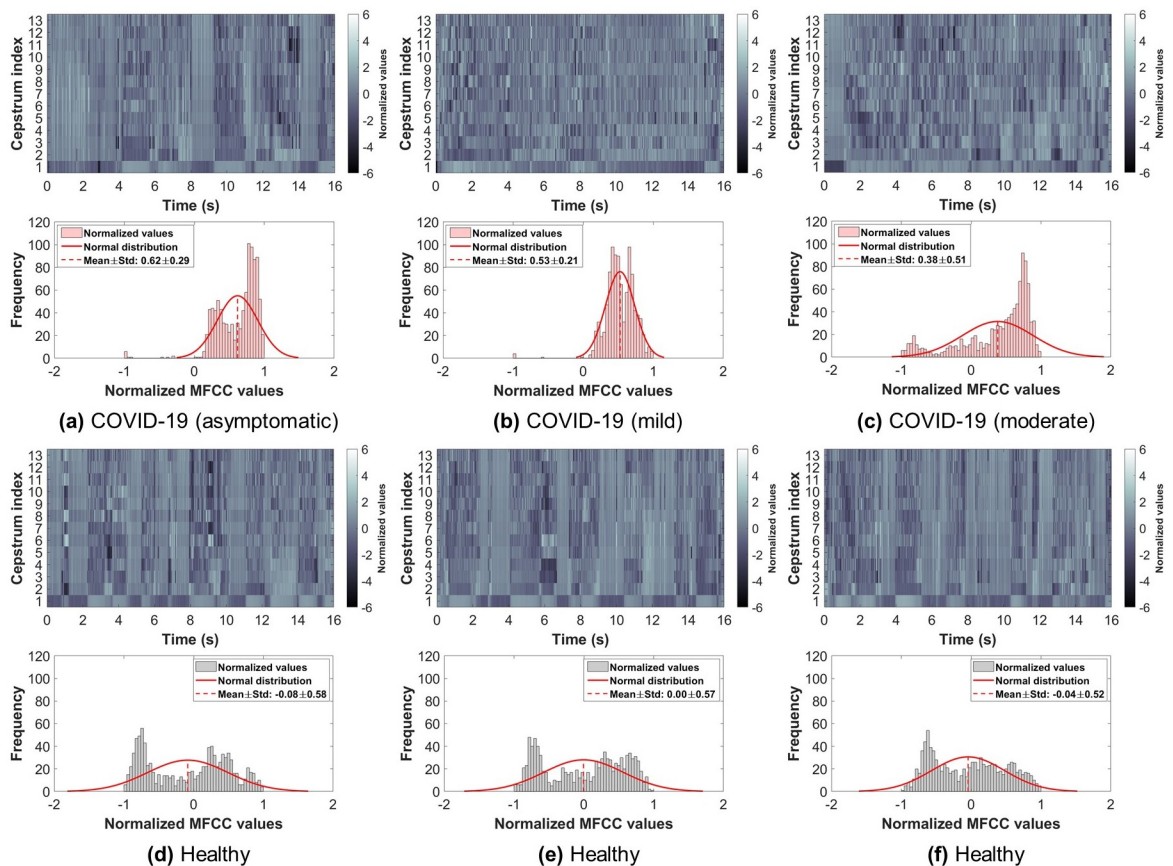

**Fig 5. Examples of the mel-frequency cepstral coefficients (MFCC) extracted from the shallow breathing dataset and illustrated as a normal distribution of summed coefficients.** Showing: (a-c) COVID-19 subjects (asymptomatic, mild, moderate), (d-f) healthy subjects.

datasets, respectively. In both datasets, the kurtosis and skewness values for COVID-19 subjects were slightly higher than healthy subjects. Furthermore, the average combined MFCC values for COVID-19 were less than those for the healthy subjects. More specifically, in the shallow breathing dataset, a kurtosis and skewness of 4.65±15.97 and 0.59±1.74 was observed for COVID-19 subjects relative to 4.47±20.66 and 00.19±1.75 for healthy subjects. On the other hand, using the deep breathing dataset, COVID-19 subjects had a kurtosis and skewness of 20.82±152.99 and 0.65±4.35 compared to lower values of 3.23±6.06 and -0.36±1.08 for healthy subjects. In addition, a statstically significant difference (using the linear regression fitting algorithm) of <0.001 was obtained between COVID-19 and healthy subjects for the combined MFCC values of the shallow and deep breathing recordings. Furthermore, the skewness of the deep breathing recordings was found significantly different with a p-value of 0.014. It is worth noting that the skewness of the shallow breathing recordings was 0.095. Moreover, the kurtosis was not significant using both datasets' recordings.

## Deep learning performance

The overall performance of the proposed deep learning model is shown in Fig 7. From the figure, the model correctly predicted 113 and 114 COVID-19 and healthy subjects, respectively, using the shallow breathing dataset out of the 120 total subjects (Fig 7(a)). In addition, only 7 COVID-19 subjects were miss-classified as healthy, whereas only 6 subjects were wrongly

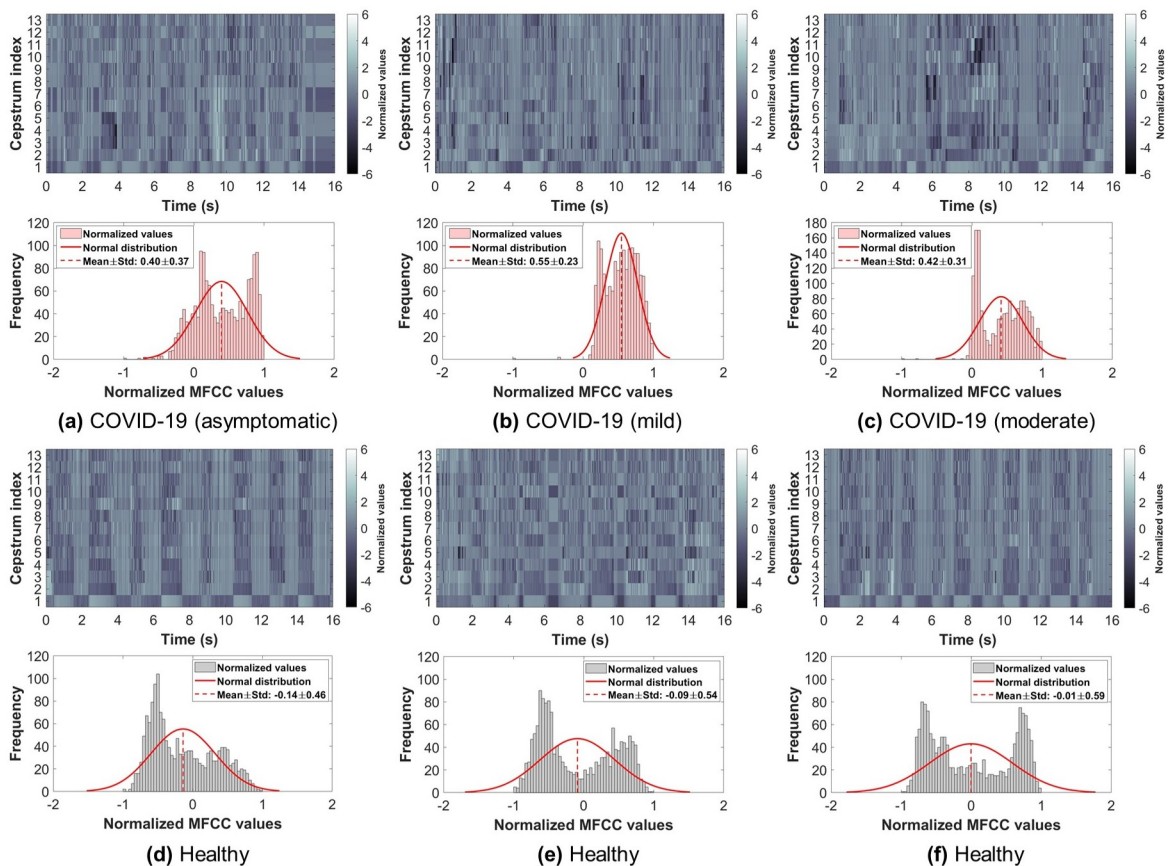

**Fig 6. Examples of the mel-frequency cepstral coefficients (MFCC) extracted from the deep breathing dataset and illustrated as a normal distribution of summed coefficients.** Showing: (a-c) COVID-19 subjects (asymptomatic, mild, moderate), (d-f) healthy subjects.

classified as carrying COVID-19. The correct predictions number was slightly lower using the deep breathing dataset with a 109 and 112 for COVID-19 and healthy subjects, respectively. In addition, wrong predictions were also slightly higher with 11 COVID-19 and 8 healthy subjects. Therefore, the confusion matrices show percentages of proportion of 94.20% and 90.80% for COVID-19 subjects using the shallow and deep datasets, respectively. On the other hand, healthy subjects had percentages of 95.00% and 93.30% for both datasets, respectively.

The evaluation metrics (Fig 7(b)) calculated from these confusion matrices returned an accuracy measure of 94.58% and 92.08% for the shallow and deep datasets, respectively.

**Table 2. Normal distribution analysis (mean±std) of the combined mel-frequency cepstral coefficients (MFCCs) using the shallow breathing dataset.**

| Category | | Normal distribution analysis | | |
|---|---|---|---|---|
| | | Combined MFCC values | Kurtosis | Skewness |
| **COVID-19** | Asymp. | -0.29±0.78 | 1.92±0.94 | 0.45±0.78 |
| | Mild | -0.24±0.70 | 5.46±18.22 | 0.61±1.95 |
| | Moderate | -0.26±0.70 | 2.26±1.85 | 0.60±0.83 |
| | Overall | -0.25±0.72 | 4.65±15.97 | 0.59±1.74 |
| **Healthy** | | -0.11±0.75 | 4.47±20.66 | 0.19±1.75 |
| **p-value** | | **<0.001** | 0.941 | 0.095 |

**Table 3. Normal distribution analysis (mean±std) of the combined mel-frequency cepstral coefficients (MFCCs) using the deep breathing dataset.**

| Category | | Normal distribution analysis | | |
|---|---|---|---|---|
| | | **Combined MFCC values** | **Kurtosis** | **Skewness** |
| **COVID-19** | Asymp. | -0.05±0.69 | 2.61±2.12 | -0.17±0.97 |
| | Mild | -0.15±0.63 | 6.63±6.51 | 0.91±4.95 |
| | Moderate | 0.04±0.58 | 2.54±1.12 | -0.15±0.96 |
| | Overall | -0.12±0.64 | 5.82±5.99 | 0.65±4.35 |
| **Healthy** | | 0.12±0.60 | 3.23±6.06 | -0.36±1.08 |
| **p-value** | | **<0.001** | 0.941 | **0.014** |

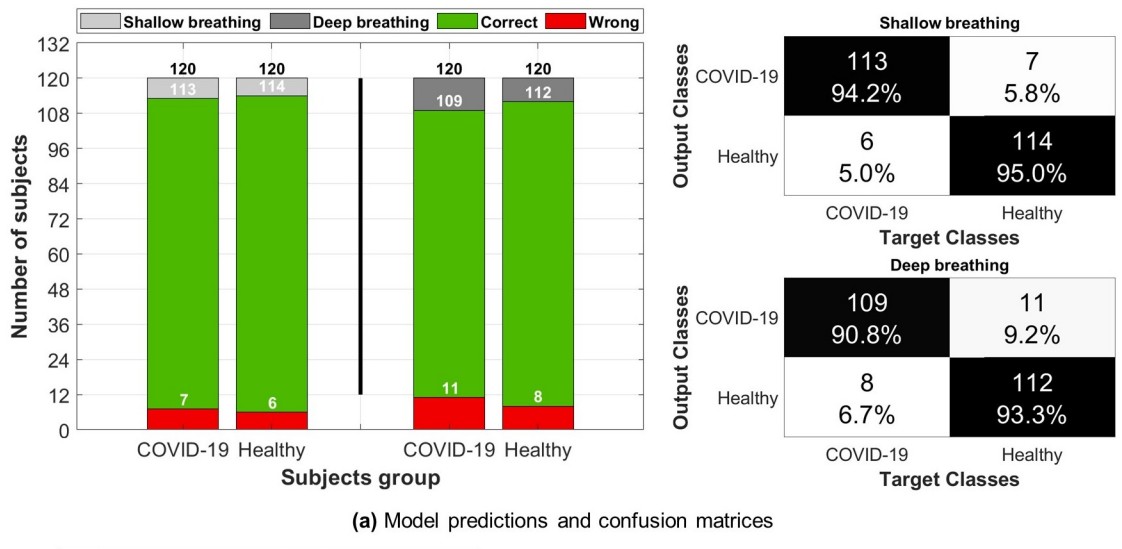

**(a)** Model predictions and confusion matrices

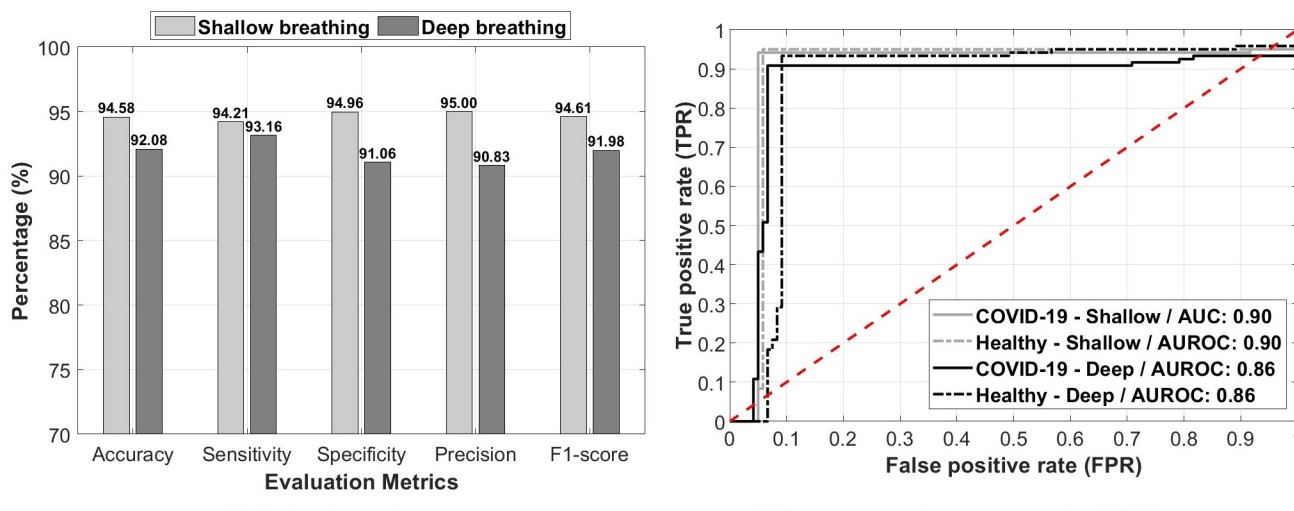

**(b)** Evaluation metrics

**(c)** Receiver operating characteristic (ROC) curves

**Fig 7. The performance of the deep learning model in predicting COVID-19 and healthy subjects using shallow and deep breathing datasets.** Showing: (a) model's predictions for both datasets and he corresponding confusion matrices, (b) evaluation metrics including accuracy, sensitivity, specificity, precision, and F1-score, (c) receiver operating characteristic (ROC) curves and corresponding area under the curve (AUROC) for COVID-19 and healthy subjects using both datasets.

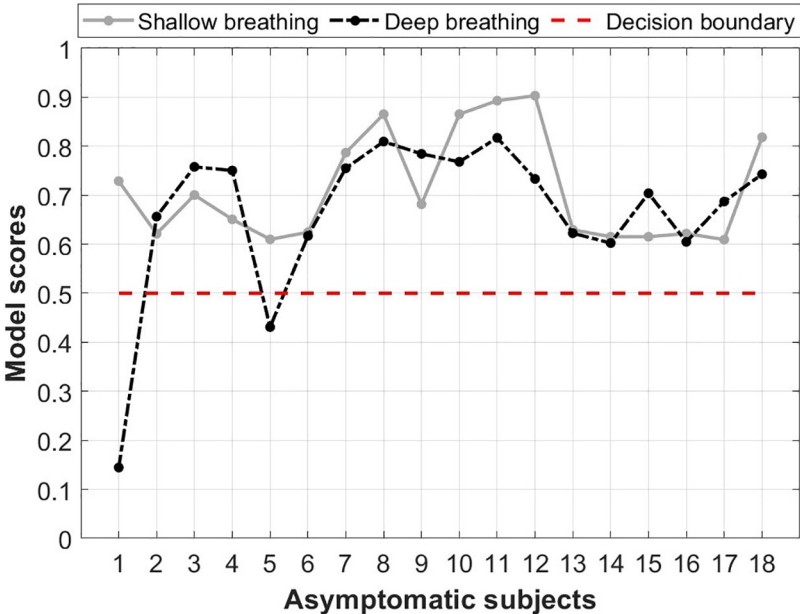

**Fig 8. Asymptomatic COVID-19 subjects' predictions based on the proposed deep learning model.** The model had a decision boundary of 0.5 to discriminate between COVID-19 and healthy subjects. The values represent a normalized probability regrading the confidence in predicting these subjects as carrying COVID-19.

Furthermore, the model had a sensitivity and specificity measures of 94.21%/94.96% for the shallow dataset and 93.16%/91.06% for the deep dataset. The precision was the highest measure obtained for the shallow dataset (95.00%), where as the deep dataset had the lowest value in the precision with a 90.83%. Lastly, the F1-score measures returned 94.61% and 91.98% for both datasets, respectively.

To analyze the AUROC, Fig 7(c) shows the ROC curves of predictions using both the shallow and deep datasets. The shallow breathing dataset had an overall AUROC of 0.90 in predicting COVID-19 and healthy subjects, whereas the deep breathing dataset had a 0.86 AUROC, which is slightly lower performance in the prediction process. Additionally, the model had high accuracy measures in predicting asymptomatic COVID-19 subjects (Fig 8). Using the shallow breathing dataset, the model had a 100.00% accuracy by predicting all subjects correctly. On the other hand, using the deep breathing dataset, the model achieved an accuracy of 88.89% by missing two asymptomatic subjects. It is worth noting that few subjects had close scores (probabilities) to 0.5 using both datasets, however, the model correctly discriminated them from healthy subjects.

## Neural network activations

Fig 9 shows the extracted neural network activations (deep-activated features) from the last layer (BiLSTM) for five examples from the COVID-19 and healthy subjects. These activations were obtained after applying the BiLSTM hidden units calculations on the flattened feature vector obtained from the CNN. The 512 hidden units are considered as the final deep-activated feature vector used to classify subjects into COVID-19 or healthy. By inspecting both COVID-19 (left column) and healthy (right column) subjects, it can be seen that the network learned successfully features that best maximize the margin between the two classes. For COVID-19, the activations were more spread all over the hidden units in a randomized manner, which

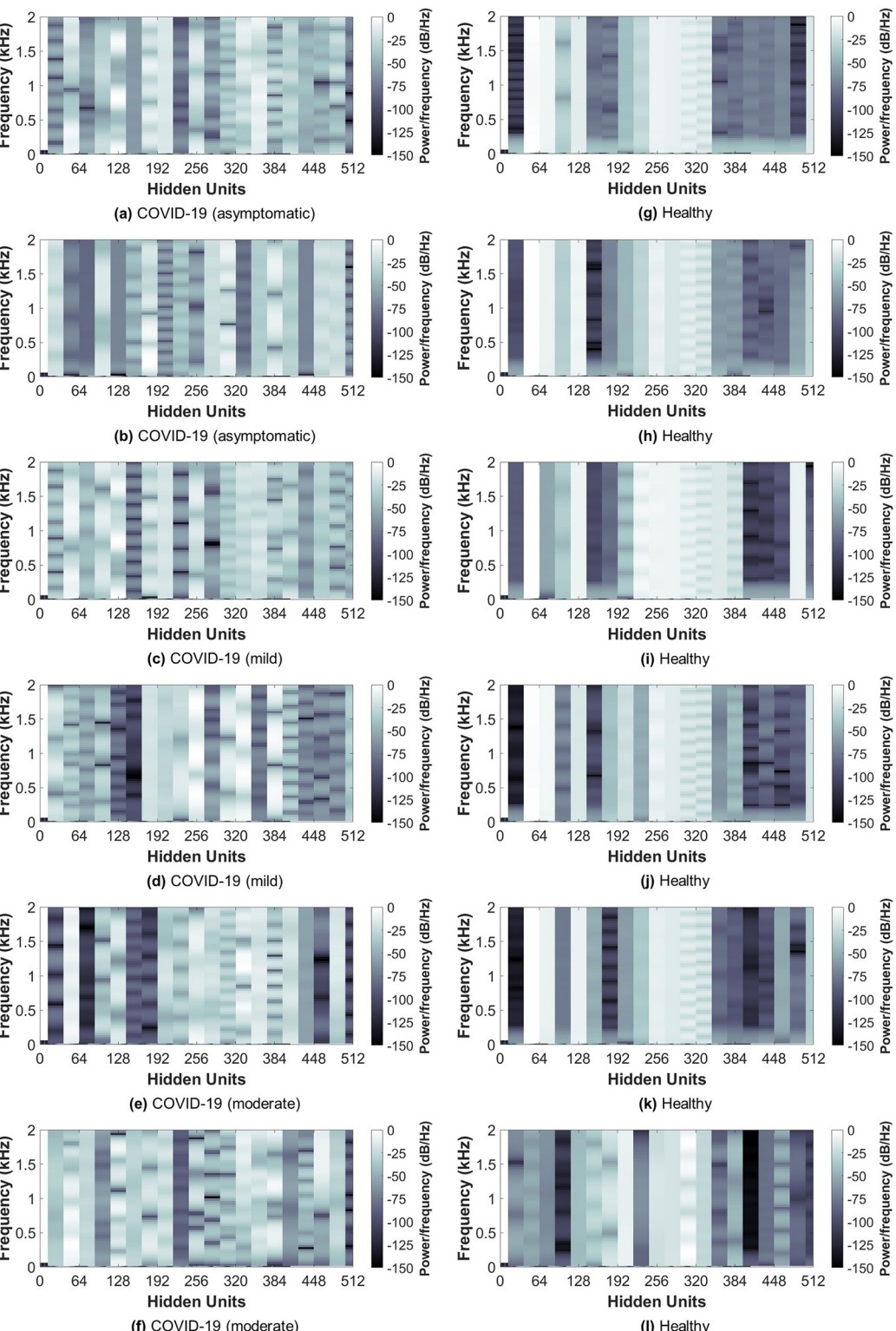

**Fig 9. Bi-directional long short-term memory (BiLSTM) network activations extracted for five examples from COVID-19 and healthy subjects using shallow breathing recordings.** The activations were extracted from the last layer (applied on the flattened convolutional neural network (CNN) features vector) of the deep learning network (BiLSTM) for five examples from COVID-19 (a-f) and healthy (g-l) subjects.

could be due to the irregular breathing patterns seen in the original breathing sounds for COVID-19 subjects (Figs 2 and 3). On the other hand, healthy subjects had a close-to-regular patterns with higher power over the 60-128 and 200-350 hidden units. Similarly, this could be due to the normal breathing patterns observed in the healthy subjects breathing recordings. The ability of the neural network to acquire such differences in both classes suggest the potential of deep learning in the discrimination through 1D breathing sounds.

## Performance relative to current state-of-art

To represent the performance of the proposed deep learning model relative to the current state-of-art studies, Table 4 shows the recent works on COVID-19 detection using machine learning and breathing/coughing recordings. The majority of studies have used coughing sounds to train deep learning networks. In addition, only two studies have utilized breathing sounds as input to the trained models [68, 69]. The only limitation in [69] is the heavy unbalance in favor of the normal subjects against COVID-19 subjects, which could have been the reason behind the high performance metrics achieved. In addition, authors in [68] use only 5 COVID-19 subjects, which does not ensure a generalized performance of deep learning networks. In contrary, the proposed study utilized a more balanced dataset with 120 COVID-19 subjects and the performance was higher than most of other studies. It is worth noting that most studies use web-based source for COVID-19 recordings, while in the proposed study breathing recordings were obtained from a smartphone app. In [69, 70], authors have used a

**Table 4. Summary table of the current state-of-art works in COVID-19 detection using machine learning and breathing/coughing recordings.**

| Study | Year | Recordings Source | Respiratory Sound | Number of Subjects | Number of Recordings | Pre-processing Steps | Trained Model | Performance |
|---|---|---|---|---|---|---|---|---|
| Bagad *et al* [70] | 2020 | Smartphone app | Cough | 2001 COVID-19 1620 healthy | 3621 | Short-term magnitude spectrogram | Convolutional neural network (ResNet-18) | Accuracy: Not applicable AUC: 0.72 |
| Laguarta *et al.* [39] | 2020 | Web-based | Cough | 2660 COVID-19 2660 healthy | 5320 | Mel-frequency cepstral coefficients (MFCC) | Convolutional neural network (ResNet-50) | Accuracy: 97.10% |
| Mohammed *et al.* [71] | 2021 | Web-based | Cough | 114 COVID-19 1388 healthy | 1502 | Spectrogram Mel spectrum, power spectrum Tonal spectrum, chroma spectrum Raw signals Mel-frequency cepstral coefficients (MFCC) | Ensemble convolutional neural network | Accuracy: 77.00% |
| Lella *et al.* [69] | 2021 | Web-based smartphone app | Cough voice breathing | 5000-6000 Subjects 300 COVID | 6000 | De-noising auto encoder (DAE) Gamm-atone frequency cepstral coefficients (GFCC) Improved Mel-frequency cepstral coefficients (IMFCC) | Convolutional neural network | Accuracy: 95.45% |
| Sait *et al.* [68] | 2021 | Electronic stethoscope | Breathing | 5 COVID-19 5 healthy | 10 | Two-dimensional (2D) Fourier transformation | Convolutional neural network (Inception-v3) | Accuracy: 80.00% |
| Manshouri *et al.* [72] | 2021 | Web-based | Cough | 7 COVID-19 9 Healthy | 16 | Mel-frequency cepstral coefficients (MFCC) Single-time Fourier transformation | Support vector machine (SVM) | Accuracy: 94.21% |
| **This study** | **2021** | **Smartphone app** | **Shallow/deep breathing** | **120 COVID-19 120 healthy** | **Total: 480 Shallow: 240 Deep: 240** | **Raw signals Mel-frequency cepstral coefficients (MFCC)** | **Convolutional neural network Bi-directional long short-term memory (CNN-BiLSTM) + Hand-crafted features** | **Accuracy: Shallow = 94.58% Accuracy: Deep = 92.08%** |

smartphone app to acquire the recordings, however, they rely on coughing sounds, which makes it even more challenging to rely only on breathing sounds (as in the proposed study) and still achieve high performance. Additionally, the proposed study uses raw breathing signals (shallow and deep) to train deep learning models with the inclusion of best 20 features extracted from the raw signals and MFCC transformations, which was not the case in any study found in literature (majority require signal transformation to 2D images).

## Discussion

This study demonstrated the importance of using deep learning for the detection of COVID-19 subjects, especially those who are asymptomatic. Furthermore, it elaborated on the significance of biological signals, such as breathing sounds, in acquiring useful information about the viral infection. Unlike the conventional lung auscultation techniques, i.e., electronic stethoscopes, to record breathing sounds, the study proposed herein utilized breathing sounds recorded via a smartphone microphone. The observations found in this study (highest accuracy: 94.58%) strongly suggest deep learning as a pre-screening tool for COVID-19 as well as an early detection technique prior to the gold standard RT-PCR assay.

### Smartphone-based breathing recordings

Although current lung auscultation techniques provide high accuracy measures in detecting respiratory diseases [73–75], it requires subjects to be present at hospitals for equipment setup and testing preparation prior to data acquisition. Furthermore, it requires the availability of an experienced person, i.e., clinician or nurse, to take data from patients and store it in a database. Therefore, utilizing a smartphone device to acquire such data allows for a faster data acquisition process from subjects or patients while at the same time, provides highly comparable and acceptable diagnostic performance. In addition, smartphone-based lung auscultation ensures a better social distancing behaviour during lock downs due to pandemics such as COVID-19, thus, it allows for a rapid and time-efficient detection of diseases despite of strong restrictions.

By visually inspecting COVID-19 and healthy subjects' breathing recordings (Figs 2 and 3), an abnormal nature was usually observed by COVID-19 subjects, while healthy subjects had a more regular pattern during breathing. This could be related to the hidden characteristics of COVID-19 contaminated within lungs and exhibited during lung inhale and exhale [35, 38, 76]. Additionally, the MFCC transformation of COVID-19 and healthy subjects' recordings returned similar observations. By quantitatively evaluating these coefficients when combined, COVID-19 subjects had a unique distribution (positively skewed) that can be easily distinguished from the one of healthy subjects. This gives an indication about the importance of further extracting the internal attributes carried not only by the recordings themselves, but rather by the additional MFC transformation of such recordings. Additionally, the asymptomatic subjects had a distribution of values that was close in shape to the distribution of healthy subjects, however, it was skewed towards the right side of the zero mean. This may be considered as a strong attribute when analyzing COVID-19 patients who do not exhibit any symptoms and thus, discriminating them easily from healthy subjects.

### Diagnosis of COVID-19 using deep learning

It is essential to be able to gain the benefit of the recent advances in AI and computerized algorithms, especially during these hard times of COVID-19 spread worldwide. Deep learning not only provides high levels of performance, it also reduces the dependency on experts, i.e., clinicians and nurses, who are now suffering in handling the pandemic due to the huge and rapidly increasing number of infected patients [77–79]. Recently, the detection of COVID-19 using

deep learning has reached high levels of accuracy through two-dimensional (2D) lung CT images [80–82]. Despite of such performance in discriminating and detecting COVID-19 subjects, CT imaging is considered high in cost and requires extra time to acquire testing data and results. Furthermore, it utilizes excessive amount of ionizing radiations (X-ray) that are usually harmful to the human body, especially for severely affected lungs. Therefore, the integration of biological sounds, as in breathing recordings, within a deep learning framework overcomes the aforementioned limitations, while at the same time provides acceptable levels of performance.

The proposed deep learning framework had high levels of accuracy (94.58%) in discriminating between COVID-19 and healthy subjects. The structure of the framework was built to ensure a simple architecture, while at the same time to provide advanced features extraction and learning mechanisms. The combination between hand-crafted features and deep-activated features allowed for maximized performance capabilities within the model, as it learns through hidden and internal attributes as well as deep structural and temporal characteristics of recordings. The high sensitivity and specificity measures (94.21% and 94.96%, respectively) obtained in this study prove the efficiency of deep learning in distinguishing COVID-19 subjects (AUROC: 0.90). Additionally, it supports the field of deep learning research on the use of respiratory signals for COVID-19 diagnostics [39, 83]. Alongside the high performance levels, it was interesting to observe a 100.00% accuracy in predicting asymptomatic COVID-19 subjects. This could enhance the detection of this viral infection at a very early stage and thus, preventing it from developing to mild and moderate conditions or spreading to other people.

Furthermore, this high performance levels were achieved through 1D signals instead of 2D images, which allowed the model to be simple and not memory exhausting. In addition, due to its simplicity and effective performance, it can be easily embedded within smartphone applications and internet-of-things tools to allow real-time and direct connectivity between the subject and family for care or healthcare authorities for services.

## Clinical relevance

The statistical observations found in this study suggested that there is a significant difference between COVID-19 and healthy subjects for the ischemic heart disease comorbidity. This matches with the current discussions in literature about the correlation between COVID-19 and cardiac dysfunctionalities [84–86]. It was found in [84] that COVID-19 could induce myocardial injury, cardiac arrhythmia, and acute coronary syndrome. In addition, several health conditions related to the respiratory system were found significant in discriminating between COVID-19 and healthy subjects including fever, cold, and cough, which are the regular symptoms observed in most COVID-19 subjects. However, it was interesting to observe that muscle pain was significant, which matches with the previous WHO reports that states a percentage of 14.8% among COVID-19 subjects studied in China [87]. It is worth noting that diarrhoea was not significant in this study, which could show no correlation between COVID-19 and its existence in subjects.

The utilization of smartphone-based breathing recordings within a deep learning framework may have the potential to provide a non-invasive, zero-cost, rapid pre-screening tool for COVID-19 in low-infected as well as servery-infected countries. Furthermore, it may be useful for countries who are not able of providing the RT-PCR test to everyone due to healthcare, economic, and political difficulties. Furthermore, instead of performing RT-PCR tests on daily or weekly basis, the proposed framework allows for easier, cost effective, and faster large-scale detection, especially for counties/areas who are putting high expenses on such tests due to logistical complications. Alongside the rapid nature of this approach, many healthcare service

could be revived significantly by decreasing the demand on clinicians or nurses. In addition, due to the ability of successfully detecting asymptomatic subjects, it can decrease the need for extra equipment and costs associated with further medication after the development of the viral infection in patients.

Clinically, it is better to have a faster connection between COVID-19 subjects and medical practitioners or health authorities to ensure continues monitoring for such cases and at the same time maintain successful contact tracing and social distancing. By embedding such approach within a smartphone applications or cloud-based networks, monitoring subjects, including those who are healthy or suspected to be carrying the virus, does not require the presence at clinics or testing points. Instead, it can be performed real-time through a direct connectivity with a medical practitioners. In addition, it can be completely done by the subject himself to self-test his condition prior to taking further steps towards the RT-PCR assay. Therefore, such approach could set an early alert to people, especially those who interacted with COVID-19 subjects or are asymptomatic, to go and further diagnose their case. Considering such mechanism in detecting COVID-19 could provide a better and well-organized approach that results in less demand for clinics and medical tests, and thus, enhances back the healthcare and economic sectors in various countries worldwide.

## Conclusion

This study suggests smartphone-based breathing sounds as a promising indicator for COVID-19 cases. It further recommends the utilization of deep learning as a pre-screening tool for such cases prior to the gold standard RT-PCR tests. The overall performance found in this study (accuracy 94.58%) in discriminating between COVID-19 and healthy subjects shows the potential of such approach. This study paves the way towards implementing deep learning in COVID-19 diagnostics by suggesting it as a rapid, time-efficient, and no-cost technique that does not violate social distancing restrictions during pandemics such as COVID-19.

## Acknowledgments

The authors would like to acknowledge project Coswara and Dr. Sriram Ganapathy for their open-access database for COVID-19 breathing recordings.

## Author Contributions

**Conceptualization:** Mohanad Alkhodari.

**Data curation:** Mohanad Alkhodari.

**Formal analysis:** Mohanad Alkhodari.

**Funding acquisition:** Ahsan H. Khandoker.

**Investigation:** Mohanad Alkhodari.

**Methodology:** Mohanad Alkhodari.

**Project administration:** Ahsan H. Khandoker.

**Resources:** Mohanad Alkhodari.

**Software:** Mohanad Alkhodari.

**Supervision:** Ahsan H. Khandoker.

**Validation:** Mohanad Alkhodari.

**Visualization:** Mohanad Alkhodari.

**Writing – original draft:** Mohanad Alkhodari.

**Writing – review & editing:** Mohanad Alkhodari, Ahsan H. Khandoker.

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
