## [Decision Letter · Decision Letter 0]

1 Nov 2021

PONE-D-21-27860Detection of COVID-19 in smartphone-based breathing recordings using CNN-BiLSTM: a pre-screening deep learning toolPLOS ONE

Dear Dr. Alkhodari,

Thank you for submitting your manuscript to PLOS ONE. After careful consideration, we feel that it has merit but does not fully meet PLOS ONE’s publication criteria as it currently stands. Therefore, we invite you to submit a revised version of the manuscript that addresses the points raised during the review process.

Specifically, the authors should demonstrate the originality of their methodology, and contribution to the research field.

We look forward to receiving your revised manuscript.

Kind regards,

Robertas Damaševičius

Academic Editor

PLOS ONE

Journal Requirements:

"This work was supported by a grant (award number: 8474000132) from the Healthcare Engineering Innovation Center (HEIC) at Khalifa University, Abu Dhabi, UAE, and by grant (award number: 29934) from the Department of Education and Knowledge (ADEK), Abu Dhabi, UAE."

"This work was supported by a grant (award number: 8474000132) from the Healthcare Engineering Innovation Center (HEIC) at Khalifa University, Abu Dhabi, UAE, and by grant (award number: 29934) from the Department of Education and Knowledge (ADEK), Abu Dhabi, UAE"

3. We note that you have stated that you will provide repository information for your data at acceptance. Should your manuscript be accepted for publication, we will hold it until you provide the relevant accession numbers or DOIs necessary to access your data. If you wish to make changes to your Data Availability statement, please describe these changes in your cover letter and we will update your Data Availability statement to reflect the information you provide

Additional Editor Comments (if provided):

The manuscript should be revised.

Reviewers' comments:

Reviewer's Responses to Questions

**Comments to the Author**

1. Is the manuscript technically sound, and do the data support the conclusions?

Reviewer #1: Yes

Reviewer #2: No

2. Has the statistical analysis been performed appropriately and rigorously? 

Reviewer #1: Yes

Reviewer #2: No

3. Have the authors made all data underlying the findings in their manuscript fully available?

Reviewer #1: Yes

Reviewer #2: Yes

4. Is the manuscript presented in an intelligible fashion and written in standard English?

Reviewer #1: Yes

Reviewer #2: Yes

5. Review Comments to the Author

Reviewer #1: In the paper, the authors propose using CNN-BiLSTM for COVID19 detection. The proposal was explained and described. In general, it is a good approach and shows great results. Some of my issues that should be improved:

1) Discuss in more detail the latest achievements in using machine learning for covid detection/classification.

2) Add a pseudocode of your proposal

3) Make a comparison with state-of-art

Reviewer #2: -This manuscript is based on handcrafted and deep features. topic of the research is interesting but unfortunately, this manuscript is lack of novelty. The CNN-BiLSTM based features extraction are not as a contributions. Also, the results section is not impressive.

I do not recommend to accept this manuscript.

6. PLOS authors have the option to publish the peer review history of their article (what does this mean?). If published, this will include your full peer review and any attached files.

Reviewer #1: No

Reviewer #2: No

---

## [Author Response · Author response to Decision Letter 0]

25 Nov 2021

REVIEWER 1

In the paper, the authors propose using CNN-BiLSTM for COVID19 detection. The proposal was explained and described. In general, it is a good approach and shows great results. Some of my issues that should be improved.

Concern #1: Discuss in more detail the latest achievements in using machine learning for covid detection/classification.

Response: The authors appreciate all concerns of this reviewer, which have significantly improved the proposed manuscript. We have added a paragraph about machine learning-related literature in COVID-19 detection using CT, X-ray, and Ultrasound imaging (See Introduction). More information about current achievements in detecting COVID-19 in breathing/coughing recordings were added as well (See Results and Table 4, Page 14) based on Concern #3.

Concern #2: Add a pseudocode of your proposal.

Response: We have added a pseudocode for the training procedure of the proposed deep learning model including feature extraction, MFCC transformation, and CNN-BiLSTM deep features extraction (See Algorithm 1, Page 6).

Concern #3: Make a comparison with state-of-art.

Response: We have created a summary table to show the current state-of-art studies in detecting COVID-19 in breathing/coughing recordings using machine learning (See Table 4, Page 14). We have added a section as well to discuss the performance of our model relative to these studies (See Results).

REVIEWER 2

This manuscript is based on handcrafted and deep features. topic of the research is interesting but unfortunately, this manuscript is lack of novelty. The CNN-BiLSTM based features extraction are not as a contribution. Also, the results section is not impressive. I do not recommend to accept this manuscript.

Response: We appreciate the concern of this reviewer about the contributions of the proposed study. This work contributes to the field of research towards developing a purely digital COVID-19 diagnostic test by applying deep learning methods to analyze breathing recordings (shallow and deep breathing). When compared to the current gold-standard PCR test, this tool would be cost-effective, safe for social distancing protocols, faster in returning results, and does not require experts nurses/clinicians to perform the assessment. Taking all this under consideration, this could allow for a better revival in the healthcare and economic sectors in many countries where COVID-19 is still hitting. In addition, the proposed study uses the raw breathing-only signals as 1D input to deep learning, which when compared to current state-of-art methods that use 2D image transformations, is considered lighter for telemedicine applications with less computational demand. Moreover, majority of studies rely on coughing recordings instead of breathing sounds, which was not the case in this work that depends on regular breathing patterns. 

We have elaborated on the performance of the trained deep learning model by extracting the BiLSTM activations. This allows for understanding the machine-based decisions when discriminating between COVID-19 and healthy subjects (See Results and Fig. 9, Page 12).

We have further improved the whole manuscript, including the results section, by adding the following,

1) Complete statistical analysis using linear regression fitting for subjects clinical information and MFCC observations for both shallow and deep breathing recordings (Tables 1, 2, and 3).

2) A new subsection in the Results section named “Patient clinical information” to discuss the observations found when statistically analyzing patient profiles.

3) Table 4 that shows a summary of the current state of art in COVID-19 detection using respiratory sounds and machine learning relative to the proposed approach.

4) A new subsection in the Results section named “Performance relative to current state-of-art” to discuss further the performance relative to literature.

5) Extra figures for shallow breathing signals and MFCC transformation (Figures 2 and 5).

6) A complete pseudocode of the proposed study for the development of a trained deep learning model (Algorithm 1).

7) Additional information in the Discussion section on the clinical observations found in this study relative to COVID-19 literature “Section Clinical relevance”.

8) Additional results after the extraction of BiLSTM network activations for COVID-19 and healthy subjects “Section Neural network activations and Fig. 9”.

9) All algorithms used to re-generate the proposed study including MFCC extraction and deep learning modelling.

We hope that the revised manuscript provides rich information to the readers. We appreciate the concern of this reviewer which have led to pushing the readability and rationale of the proposed work, and we appreciate their re-consideration of our manuscript in its revised form.

---

## [Decision Letter · Decision Letter 1]

13 Dec 2021

PONE-D-21-27860R1Detection of COVID-19 in smartphone-based breathing recordings: a pre-screening deep learning toolPLOS ONE

Dear Dr. Alkhodari,

Thank you for submitting your manuscript to PLOS ONE. After careful consideration, we feel that it has merit but does not fully meet PLOS ONE’s publication criteria as it currently stands. Therefore, we invite you to submit a revised version of the manuscript that addresses the points raised during the review process.

Specifically, the description of the methodology needs to be improved, and the comparison of results with previous works must be presented.

We look forward to receiving your revised manuscript.

Kind regards,

Robertas Damaševičius

Academic Editor

PLOS ONE

Journal Requirements:

Reviewers' comments:

Reviewer's Responses to Questions

**Comments to the Author**

1. If the authors have adequately addressed your comments raised in a previous round of review and you feel that this manuscript is now acceptable for publication, you may indicate that here to bypass the “Comments to the Author” section, enter your conflict of interest statement in the “Confidential to Editor” section, and submit your "Accept" recommendation.

Reviewer #1: All comments have been addressed

Reviewer #2: (No Response)

2. Is the manuscript technically sound, and do the data support the conclusions?

Reviewer #1: Yes

Reviewer #2: (No Response)

3. Has the statistical analysis been performed appropriately and rigorously? 

Reviewer #1: Yes

Reviewer #2: (No Response)

4. Have the authors made all data underlying the findings in their manuscript fully available?

Reviewer #1: Yes

Reviewer #2: (No Response)

5. Is the manuscript presented in an intelligible fashion and written in standard English?

Reviewer #1: Yes

Reviewer #2: (No Response)

6. Review Comments to the Author

Reviewer #1: In my opinion, the paper is ready for publication. The authors improved the paper to all comments.

Reviewer #2: This manuscript is well revised; however, a minor revision is required to improve it further.

1) Deep learning is an important research topic, so how useful for COVID? add some related methods in the related work such as; i) COVID-19 Case Recognition from Chest CT Images by Deep Learning, Entropy-Controlled Firefly Optimization, and Parallel Feature Fusion; ii) Deep Rank-Based Average Pooling Network for Covid-19 Recognition; iii) Screening of COVID-19 Patients Using Deep Learning And IoT Framework.

2) The related work section should be improve. i) Pseudo Zernike Moment and Deep Stacked Sparse Autoencoder for COVID-19 Diagnosis; ii) COVID19 Classification Using CT Images Via Ensembles of Deep Learning Models; iii) Prediction of COVID-19-pneumonia based on selected deep features and one class kernel extreme learning machine; iv) A novel framework for rapid diagnosis of COVID-19 on computed tomography scans

3) Algorithm 1 should be written in the form of proper algorithm like mathematical.

4) What are the difference here among spectral and simple entropy?

5) How deep activated features are computed? add some point to point description.

6) No need of eq. 13-17, these measures are well known. just add the name and ref.

7) Th comparison is not added. If you are not interested to add comparison then add a statistical analysis. You can reffer the following work:

i) Intelligent Fusion-Assisted Skin Lesion Localization and Classification for Smart Healthcare

ii) A two‐stream deep neural network‐based intelligent system for complex skin cancer types classification

7. PLOS authors have the option to publish the peer review history of their article (what does this mean?). If published, this will include your full peer review and any attached files.

Reviewer #1: No

Reviewer #2: No

---

## [Author Response · Author response to Decision Letter 1]

23 Dec 2021

REVIEWER 1

In my opinion, the paper is ready for publication. The authors improved the paper to all comments

REVIEWER 2

This manuscript is well revised; however, a minor revision is required to improve it further

Concern #1: Deep learning is an important research topic, so how useful for COVID? add some related methods in the related work such as; i) COVID-19 Case Recognition from Chest CT Images by Deep Learning, Entropy-Controlled Firefly Optimization, and Parallel Feature Fusion; ii) Deep Rank-Based Average Pooling Network for Covid-19 Recognition; iii) Screening of COVID-19 Patients Using Deep Learning and IoT Framework.

Response: We appreciate the re-consideration of our manuscript by this reviewer. Accordingly, we have added the suggested works to the introduction and literature review of the revised manuscript (See Section Introduction, Page 2).

Concern #2: The related work section should be improved. i) Pseudo Zernike Moment and Deep Stacked Sparse Autoencoder for COVID-19 Diagnosis; ii) COVID19 Classification Using CT Images Via Ensembles of Deep Learning Models; iii) Prediction of COVID-19-pneumonia based on selected deep features and one class kernel extreme learning machine; iv) A novel framework for rapid diagnosis of COVID-19 on computed tomography scans

Response: We have added the suggested works to the introduction and literature review of the revised manuscript (See Section Introduction, Page 2).

Concern #3: Algorithm 1 should be written in the form of proper algorithm like mathematical

Response: We preferred to keep the algorithm simple and easy to follow with less mathematical derivations to prevent confusions when re-doing the study by other researchers in future. However, we have kept all mathematical equations and derivations of features in text with proper indexing within the Algorithm. 

Concern #4: What are the difference here among spectral and simple entropy

Response: Briefly, in sample entropy, the signal is analyzed in time and phase domains. On the other hand, spectral entropy analyzes the frequency spectrum of the signal. Therefore, we end up by extracting features from time and frequency domains. We have added more information to the descriptions of both features in the revised manuscript (See Section Hand-crafted features, page 7).

Concern #5: How deep activated features are computed? add some point to point description

Response: We compute deep activated features from the last layer of the network, which is the BiLSTM. These features are learned by the network as the second part in the network after the extracted CNN features. To compute these activations, we use function activations() in MATLAB inputting the trained network, the selected data (breathing recording), and the chosen features layer (BiLSTM). More information was added to the manuscript (See Section BiLSTM activations, page 9).

Concern #6: No need of eq. 13-17, these measures are well known. just add the name and ref

Response: We have updated the revised manuscript accordingly by removing the equations and adding a suitable reference (See Section Performance evaluation, page 10)

Concern #7: The comparison is not added. If you are not interested to add comparison then add a statistical analysis. You can refer the following work: i) Intelligent Fusion-Assisted Skin Lesion Localization and Classification for Smart Healthcare ii) A two‐stream deep neural network‐based intelligent system for complex skin cancer types classification

Response: We provided a complete comparison table of our dataset, methodology, and performance with other research works in the same field (AI and breathing sounds) after the suggestions of the first reviewer (See Table 4, Section Performance relative to current state-of-art, page 14). The table shows the performance of using shallow breathing and deep breathing recordings (accuracy in %) versus current state-of-art studies. In addition, the comparison between shallow breathing and deep breathing recordings are provided in Figures 7 and 8. We hope that the inclusion of the comparison table and figures satisfies this concern for the reviewer.

---

## [Decision Letter · Decision Letter 2]

26 Dec 2021

Detection of COVID-19 in smartphone-based breathing recordings: a pre-screening deep learning tool

PONE-D-21-27860R2

Dear Dr. Alkhodari,

We’re pleased to inform you that your manuscript has been judged scientifically suitable for publication and will be formally accepted for publication once it meets all outstanding technical requirements.

Kind regards,

Robertas Damaševičius

Academic Editor

PLOS ONE

Additional Editor Comments (optional):

Reviewers' comments:

Reviewer's Responses to Questions

**Comments to the Author**

1. If the authors have adequately addressed your comments raised in a previous round of review and you feel that this manuscript is now acceptable for publication, you may indicate that here to bypass the “Comments to the Author” section, enter your conflict of interest statement in the “Confidential to Editor” section, and submit your "Accept" recommendation.

Reviewer #2: (No Response)

2. Is the manuscript technically sound, and do the data support the conclusions?

Reviewer #2: (No Response)

3. Has the statistical analysis been performed appropriately and rigorously? 

Reviewer #2: (No Response)

4. Have the authors made all data underlying the findings in their manuscript fully available?

Reviewer #2: (No Response)

5. Is the manuscript presented in an intelligible fashion and written in standard English?

Reviewer #2: (No Response)

6. Review Comments to the Author

Reviewer #2: Authors successfully addressed all my comments and now this manuscript is ready for the publication.

7. PLOS authors have the option to publish the peer review history of their article (what does this mean?). If published, this will include your full peer review and any attached files.

Reviewer #2: No

---

## [Editor Report · Acceptance letter]

3 Jan 2022

PONE-D-21-27860R2 

Detection of COVID-19 in smartphone-based breathing recordings: a pre-screening deep learning tool 

Dear Dr. Alkhodari:

I'm pleased to inform you that your manuscript has been deemed suitable for publication in PLOS ONE. Congratulations! Your manuscript is now with our production department. 

Kind regards, 

on behalf of

Professor Robertas Damaševičius 

Academic Editor

PLOS ONE